# Extending R2RML to a source-independent mapping language for typed literals

Aparna Nayak , Bojan Božić , and
Luca Longo

ML-Labs, Technological University Dublin, Dublin, Republic of Ireland
{aparna.nayak, bojan.bozic, luca.longo}@tudublin.ie

**Abstract.** Linked data is often generated from raw data with help of mapping languages. However, such techniques do not allow complex data transformations which either can be implemented as custom solutions or separated from the mapping process. In this paper, we propose an approach of separating complex data transformations from the mapping process that can be still reusable across the systems. Complex data transformations include the entailment of (i) language tag and (ii) datatype that are present at the data source. The proposed method also includes inferring missing datatype information. We extended R2RML-F to handle data transformations. Our approach is validated on the test cases specified by the RDF mapping language (RML). The proposed method considers data in the form of JSON, thus making the system interoperable and reusable.

**Keywords:** Knowledge graphs · Linked data · Mapping language · Typed literals

## 1 Introduction

The meaning of the term 'Knowledge Graph' has evolved with the announcement of the Google Knowledge Graph. It has positively impacted the semantic uplifting of different structures of data appearing in heterogeneous formats. Therefore, various approaches have been proposed to generate knowledge graphs from existing (semi-)structured data. One of the approaches involves the use of direct mappings from (semi-)structured data to the Resource Description Framework (RDF) format[1]. These mappings are in the form of rules. Detaching the rules renders them to be interoperable between implementations while the systems that process those rules are independent of the use-case. The W3C standard Relational Database (RDB) to RDF Mapping Language(R2RML) is a widely used language for specifying mappings needed to generate RDF datasets from relational databases [2]. Uplifting data from any form to RDF requires the selection

---

[1] https://www.w3.org/RDF/
[2] https://www.w3.org/TR/r2rml/

of appropriate vocabulary. It helps to achieve the objective of the Semantic Web that is to connect all the entities for reuse across application, enterprise, and community.

RDF stores the data in the form of triples (subject, predicate, and object). Subject and predicate in RDF are identified by Internationalized Resource Identifier (IRI). Objects are represented in the form of literals which can be a string or a number. Objects having literals can be annotated with optional type information, such as datatype and optional language tag to describe the language used to denote an object. A datatype is a classification of data that describes types of RDF literals and are adopted from XML Schema [3]. XML schema supports two classes of datatypes: simple and complex. Simple datatypes can be primitive or derived. Each simple datatype can be uniquely addressed via a Uniform Resource Identifier (URI). Annotating RDF literal with language helps to match and integrate RDF documents[4]. Language tags help to identify the language of the written content. The use of correct language tag helps in accessibility, translation tools, page rendering, and search.

Many dedicated mapping languages were proposed for RDF mapping. However, to the best of our knowledge, a mapping language that supports the annotation of language tag and datatype that is already present in the dataset as a separate tuple doesn't exist. In this paper, we annotate literals with an appropriate datatype and language tags that are already provided in the (semi-)structured dataset. In the case of the missing datatype, it is inferred by considering core datatypes. We aim to answer the following research questions:

*To what extent can the R2RML-F mapping technique be extended to support annotation of datataype and language tag, by incorporating a separate complex data transformations from mapping process?*

Answer to the above mentioned research question is explored in section 3. R2RML is W3C standardized mapping language. It supports typed literals. However, typed literals consider only hard-coded value for the datatype or language tag. The requirement is to override the behaviour of a typed literal creation. The proposed solution annotates existing language tag and datatype to the literal.

The remaining part of this paper is structured as follows: Section 2, outlines the literature work on mapping languages and other functionalities supported by it. Section 3, gives a detailed explanation of the implementation of language tag and datatype entailment. Section 4, discusses the result with multiple examples. Finally, section 5 covers the conclusion.

## 2   Literature review

R2RML is a mapping language to express customized mappings from relational databases to RDF datasets. These mappings allow us to represent the relational

---

[3] https://www.w3.org/TR/xmlschema-2/

database that conforms to schema in RDF format. The main goal of R2RML is to directly map relational databases to RDF datasets. An advantage of R2RML over direct mapping is, a mapping author can define highly customized views over relational data. R2RML supports 'Transformation function' [5], that allows to represent a literal value in a different (syntactic) representation in RDF. However, these transformation considers underlying database technology that allows required conversion. R2RML-F [1], is an extended version that captures the function in mapping. It also allows to uplift of CSV files into RDF [6]. The functions are executed with the help of ECMAScript, hence absence of functionality by underlying technology does not make a difference to the core work.

On the other hand, R2RML can be extended altogether to add more functionality such as support of different sources, serialization formats etc. One such mapping language is RML. It is defined as a superset of the mapping language R2RML, that aims to extend applicability and score of R2RML [3]. RML is a generic mapping language defined defined to express customized mapping rules from heterogeneous data structures and serializations to the RDF data model [2]. At this stage, however, we choose to extend R2RML-F [1] to validate our ideas and will consider RML at a later stage.

## 3   Implementation

Complex data transformations remain out of scope for mapping languages. One such concern is annotation of literal with language tag and datatype. In RDF, typed literal comprise a literal value and a datatype or a language tag. Typed literal allows annotation of literal with datatype and language tag. Listing 1 shows the syntax to generate typed literal in Apache Jena framework [4]. However, this function supports only hard-coded values for language tags and datatype.

**Listing 1.** Creation of typed literal

```
model.createTypedLiteral(value, datatype)
```

The proposed method is an extension of R2RML-F [1]. R2RML-F extends R2RML to uplift CSV into RDF format. Along side, it also supports user-defined function. R2RML-F requires a function name and a function body to declare a function. The function call contains function name and parameters that has to be processed. Each function must have exactly one function name and exactly one function body. The function body is processed to compute the output and value is returned to the called function. Any error with the function output will be sent back to the user.

---

[4] https://jena.apache.org/

Two special cases are added to the existing R2RML-F functionality. First case is annotation of datatype and the second case is annotation of language tag. Function in R2RML-F is defined as shown in listing 2. In the example,'attachdt' is a function name. Function body will get executed and a value is returned.

```
<#Attachdt>
rrf:functionName ''attachdt" ;
rrf:functionBody ''''''
//Omitted
"""  ;
.
```

**Listing 2.** Function declaration in R2RML-F

The datatype and language tag that are present in the data source is rendered as a column. Therefore, the column that holds literal values and the column that holds datatype value must be specified in parameter bindings. Listing 3 depicts a function call from R2RML object map. All the numbers that belong to column 'NUM' are annotated with corresponding values present in 'DT' column.

```
rrf:functionCall  [
rrf:function <#Attachdt> ;
rrf:parameterBindings (
    [rr:column ''NUM"]
    [rr:column ''DT"]
) ;
]  .
```

**Listing 3.** Function in a Predicate Object Map

When the function name is neither 'attachdt' nor 'attachlg' the system executes the body of the function and returns appropriate value. Additionally, data is preprocessed to bring the data in required format before converting it into RDF format. The steps followed in proprocessing are discussed in the following section 4.

## 4   Experiment and Result

We demonstrate and evaluate our approach with test cases specified by RML [5]. RML test cases are designed to verify the mapping language and there are no test cases to support annotation of datatype and language tag. Test cases are modified to support language tag and datatype. Also, all the use cases provided by knowledge graph construction workshop [6] for datatype and language tags are considered along with modified RML test cases.

JSON data considered is pre-processed to get all the fields in required format of mapping. The pre-processing steps include the steps as shown in algorithm 4.

---

[5] https://rml.io/test-cases/
[6] https://github.com/kg-construct/mapping-challenges/tree/main/challenges

The algorithm considers JSON data as input, preprocesses the data to required format and outputs the comma separated values.

---

**Algorithm 1** Data preprocessing

---

**Result:** Comma separated values of data
Read JSON data.
  Initialize a stack.
  **while** *all key-value pairs* **do**
  |    Push the key value pair into the stack.
  |    **if** *key=='dt'* **then**
  |    |   Pop the value.
  |    |   **if** *value!='xsd:nnn'* **then**
  |    |   |   Update the value to 'xsd:nnn' format.
  |    |   **end**
  |    |   **if** *value==' '* **then**
  |    |   |   Predict the type based on recent pushed item.
  |    |   **end**
  |    **end**
  |    Push the value into the stack.
  |     Separate key-value pairs into two different list.
  |     Identify repeated patterns in key list to calculate total number of rows and columns.
  |     Generate dataframe based in total number of rows and columns.
  |     **if** *languageTag header* **then**
  |     |   Replace all complete language words into tag.
  |     **end**
  |     Create a comma separated values using dataframe.
**end**

---

The value of datatype that we are interested is in the form of "xsd:nnn". Here, nnn can be any RDF-compatible core datatype[7]. Missing datatypes are predicted either based on regular expression or based on JSON type. For example, datatype value for "xsd:integer" can be in the form of int, integer, or "http://www.w3.org/2001/XMLSchema#integer" irrespective of the sensitivity of the case. All the forms are finally converted into "xsd:integer".

Assuming that no missing keys and values in the JSON objects, repeated pattern of key is identified to create a header for CSV data. All the values are converted into dataframe based on total number of elements identified in header. Language tags are also updated based on the following two scenarios.

– **Language tag is given:** Value that language tag considers are in the form of 'en' for english, 'fr' for french, 'ga' for irish and so on. These values can be directly substituted to create typed literals.

---

[7] https://www.w3.org/TR/rdf11-concepts/

- **Language is given:** When the language is given in the form of full word, it is replaced with corresponding tag. All the languages and its corresponding tags are scraped from W3 schools[8].

Once we have all the datatype and language tag in required format the entire dataframe in written as comma separated values. Mapping for each test case is written and converted as per extended R2RML-F. Table 1 shows the different scenarios that have been considered for evaluation process. The code is made publicly available at github repository[9].

**Table 1.** Evaluation scenarios of extended R2RML-F

| Scenario | Language tag | Datatype |
| --- | --- | --- |
| Single object | Yes | Missing datatype |
| Single object | Yes | Yes |
| Single object | Yes | No |
| Single object | No | Missing datatype |
| Single object | No | Yes |
| Single object | No | No |
| Multiple objects | Yes | Missing datatype |
| Multiple objects | Yes | Yes |
| Multiple objects | Yes | No |
| Multiple objects | No | Missing datatype |
| Multiple objects | No | Yes |
| Multiple objects | No | No |

One special scenario where languages are described as separate JSON object is also implemented, however this requires JSON head to be named as "languages". Some of the limitations of the research are

- In case of multiple object scenario, total items in each object should be of same length.
- Keys in the JSON should be a single word.
- Keys and values can not be null values.
- Headers of the datatype column and language column should be declared as 'dt' and 'lang'.

The preprocessed data later used to create RDF triples. This is done with the help of modified R2RML-F. R2RML-F engine returns typed literals when the function are named as "attachdt" or "attachlg". Each input file requires a mapping file associated with it. Based on the customized mapping, data is converted into RDF triple format. Generated RDF triples are validated on easyrdf [10]. This helps to understand the generated data is in proper RDF format.

---

[8] https://www.w3schools.com/tags/ref_language_codes.asp
[9] https://github.com/aparnanayakn/r2rmlpreprocess
[10] https://www.easyrdf.org/

The listing 4 shows sample input considered, listing 5 shows preprocessed output, and listing 6 shows data in RDF TURTLE format. The example includes various scenarios such as missing datatype, language tag as a complete language name, multiple objects. Missing datatypes are inferred, language tags are converted into required format to generate RDF that depicts annotated data.

```
{
    ``persons": [
        {
            ``firstname": ``John", ``lastname": ``Doe",
            ``lang": ``english", ``num": 3, ``dt": `` "
        },
        {
            ``firstname": ``Jane", ``lastname": ``Smith",
            ``lang": ``fr", ``num": ``3.14", ``dt" : `` "
        }
    ]
}
```

**Listing 4.** Sample input

```
firstname , lastname , lang , num , dt
John , Doe , en , 3.0 , xsd : integer
Jane , Smith , fr , 3.14 , xsd : string
```

**Listing 5.** Preprocessed output

```
<http :// data . example . org /sw/ John>
        a          <http ://www. example . org / ont#Person> ;
        <http ://www. example . org / ont#FNAME>
                ``John" ;
        <http ://www. example . org / ont#LNAME>
                ``Doe"@en ;
        <http ://www. example . org / ont#NUM>
                ``3.0"^^<xsd : integer > .

<http :// data . example . org /sw/ Jane>
        a          <http ://www. example . org / ont#Person> ;
        <http ://www. example . org / ont#FNAME>
                ``Jane" ;
        <http ://www. example . org / ont#LNAME>
                ``Smith"@fr ;
        <http ://www. example . org / ont#NUM>
                ``3.14"^^<xsd : string > .
```

**Listing 6.** Output in RDF TURTLE format

## 5   Conclusion

In this paper, we presented a simple approach to map JSON data sources into RDF and to create typed literals using extended R2RML-F. Proposed method follows the principle separating complex data transformations from mapping process. Our method efficiently solves the research questions discussed in section 1. The proposed model is evaluated using RML test cases by adding datatype and language tag in the input file. The model's extensibility is self-evident as the whole solution is separated from mapping process. R2RML-F treats function calls as a term map. These mapping function helps to create typed literals in RDF.

Future work includes creating typed literals using heterogeneous data structures and thus using RML, along with additional experiments to validate our findings and developing additional scenarios.

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
