# OpenReview forum: "Extending R2RML to a source-independent mapping language for typed literals"
_eswc-conferences.org/ESWC/2021/Workshop/KGCW — Submitted to KGCW 2021_

### Official Review · ~Ben_De_Meester1 · 2021-04-07
**Interesting approach that is not sufficiently argumented**

**Rating:** 4
**Confidence:** 5

**Review:**

This paper discusses an algorithm and implementation to support the generation of languagetagged and datatyped terms using R2RML-F and a preprocessing step. For me, the paper does not provide much insights into the reasoning of the approach, nor the argumentation of the algorithm.
Also comparison with related work is lacking. It might spark interesting discussion during the workshop, but for that, I would expect more discussion points to be raised in the paper itself, so that people not attenting the workshop could also benefit from the work.

### General remark

- The paper doesn't feel well aligned with a workshop in a semantic web conference: too much generic background
- The approach is not aligned -- nor discussed -- with respect to the KGC workshop's mapping challenges
  - e.g., RML LanguageMap is discussed in those challenges and given as possible solution: https://github.com/kg-construct/mapping-challenges/tree/main/challenges/language-map/solutions . I would have expected a discussion on how this relates to that solution.
- Argumentation of the algorithms is limited
  - Little argumentation is given about the actual datatype detection algorithm. It seems reasonable, but it's hard to understand why exactly those choices were made: is this based on a project, partner requirements, ad hoc decisions? How this algorithm came to be seems to be the most important, without argumentation, it feels like this application of a specific algorithm on top of R2RML-F has little added value for other use cases. Also, "Predict the type based on recent pushed item" is very underspecified in my opinion :). However the code is online, for which I really commend the authors, so I could actually see for myself ;).
  - The language detection algorithm is better argumented (although one could argue that there exit more authoritative sources than W3schools), however, I again wonder why exactly these choices were made (for example, not thought on internationalization was given, so it seems). This could be fine, but it would be nice to clearly state the upsides and downsides
- Limited discussion
  - One important limitation that should also be mentioned in my opinion, is that your approach does not allow to generate multiple predicateObjectMaps with different datatypes.
  - I'm a bit disappointed that the conclusion did not discuss the actual approach: you state you follow the principle "separating complex data transformations from mapping process", but you don't argument why, and what are the upsides/downsides (mostly because data transformations are already integrated in the mapping process using R2RML-F: why exactly did you not take the same approach?)

### Smaller remarks

- I don't have the feeling the titel covers the load that well
- 1. Introduction
  - "Objects are represented in the form of literals which can be a string or a number. Objects having literals can be annotated with optional type information, such as datatype and optional language tag to describe the language used to denote an object" -> that's misrepresenting the truth
- 2. Related work
  - I'm a bit disappointed more recent related work is not at least mentioned (I don't think a workshop paper needs a thorough SOTA review though), e.g., your approach seems quite related architecture-wise with FunMap https://www.researchgate.net/publication/346220361_FunMap_Efficient_Execution_of_Functional_Mappings_for_Knowledge_Graph_Creation
- 3. Implementation
  - "However, this function supports only hard-coded values for language tags and datatype." -> I don't understand how this argument related to R2RML-F. Moreover, the Listing doesn't argument anything in particular about hard-coded dataypes. In fact, it seems this is inconsistent with the actual Jena documentation https://jena.apache.org/documentation/javadoc/jena/org/apache/jena/rdf/model/Model.html#createLiteral(java.lang.String,java.lang.String)
- please make sure all urls are correctly rendered in the PDF, I almost thought there was a broken link in the PDF because one URL I could not properly copy and got https://www.w3schools.com/tags/reflanguagecodes.asp instead of https://www.w3schools.com/tags/ref_language_codes.asp

---

### Official Review · ~Ana_Iglesias-Molina2 · 2021-04-14
**Extending R2RML to a source-independent mapping language for typed literals**

**Rating:** 5
**Confidence:** 3

**Review:**

This paper proposes an extension of R2RML-F to map dynamically datatypes and language tags using functions. The proposal is implemented and tested with modified RML test cases.

I find the proposal interesting, since authors present a solution to some of the KGC Community Group mapping challenges based in the used of functions, instead of modifying the syntax of the language itself. However, I find some issues. The first one is related to the title of the paper, that in my opinion does not quite represent correctly the content of the paper and can be misleading. Authors propose extending R2RML-F, not R2RML, and the extension is limited to dynamic datatype and language tags.

I find the justification of why authors chose to extend R2RML-F insufficient. The related work does not mention current languages that do implement functions (or contemplate their use with the Function Ontology). It seem the decision is made upon the assumption that there are no other languages that use functions. I missed a more detailed motivation here.

I also missed description and references to the actual KGC-CG mapping challenges authors address ([https://github.com/kg-construct/mapping-challenges/tree/main/challenges](https://github.com/kg-construct/mapping-challenges/tree/main/challenges)) when presenting the extension, before Section 4.

The idea of using test cases to evaluate the approach is OK, but authors find the need of modifying them to actually test their proposal, and the original language for which the test cases were proposed (RML) is not the one authors are extending (R2RML-F). That makes me think that maybe evaluating the proposal with a real use-case may have been equally valid and have given the paper a more clear motivation.

Minor issues:
* In the abstract, last sentence: "The proposed method considers data in the form of JSON, thus making the system interoperable and reusable." I don't see the causality between the first part of the sentence and the second.
* In Listing 5, I find confusing why the datatype of "3.14" is xsd:string and not, for instance, xsd:decimal
* Typos:
    * Page 3, line 7: "that captures **functions in mappings**"
    * Page 3, line 8: "**uplift CSV** files", without "of" between "uplift" and "CSV"
    * Page 5, first line after Algorithm 1: "that we are interested **in** is", add "in"
    * Page 6, line 10 after Table 1: "The preprocessed data **is** later used", add "is"
    * Page 8, line 1: "In this paper, we **present**"
    * Page 8, line 2: "**The** proposed", add "The"

---

### Official Review · ~Souripriya_Das1 · 2021-04-15
**addresses the important problem of allowing dynamic datatypes and language-tags when mapping JSON data to RDF, but the approach could have been presented much better**

**Rating:** 5
**Confidence:** 3

**Review:**

This paper addresses the important problem of allowing data types and language-tags in RDF literals, produced by mapping of JSON data, to be determined dynamically. The approach involves two phases: in the first, a preprocessing algorithm is used on a JSON document to produce a CSV containing column(s) for datatype and/or language-tag information; in the second, two special functions are added to R2RML-F, namely attachdt and attachlg, that take as parameters a pair of column names -- the first parameter is for the value of the literal and the second is for the literal type or language-tag -- and return a typed-literal or a literal with a language-tag, respectively.

While the second phase of the approach is quite simple because it expects the literal type and/or language-tags to have been already determined and placed in the CSV, the preprocessing phase is not so straightforward. The algorithm for preprocessing could have been presented in much more detail, esp. w.r.t. determination of the literal types, including illustrations with examples to explain what choices make sense for different types of inputs. For example, it was a bit surprising to see "3.14" being marked as xsd:string instead of say xsd:decimal. An explanation would help clear any confusion.

The organization of the paper leaves a lot of room for improvement. For example, it would have been helpful to the reader to 1) see background materials clearly separated from the description of the approach; 2) including simple but illustrative examples in the introduction; 3) after the long example in Section 4, identify how the approach would still work if some of the specific assumptions used in the example (for simplicity) were relaxed to go to more general cases. Also, there were many typos in the paper (for example, "mapping language dened dened to express customized mapping")  -- a careful proofreading would improve its readability.

---

### Official Review · ~Guohui_Xiao1 · 2021-04-18
**Interesting work but not ready for publication**

**Rating:** 4
**Confidence:** 4

**Review:**

The paper proposed an extension of the R2RML-F mapping language to specify the datatype and language tags. This proposed extension is necessary and useful, however, the manuscript did not describe the contribution properly, and there are a number of issues in the writing.

Section 2. It is not clear why the extension is JSON-specific. In principle, the same problem also happens in other formats, e.g. CSV and even relational databases.

Section 3. Implementation. It is not clear what has been implemented. The implementation only provides the functions in Listings 2 and 3.  Is there a system that actually implemented the proposed method?

Section 4. The experiment seems too limited. It looks like only one example with the test case.

Minor issue: please use a better fix-width font for typing setting the Listing environments. Use e.g. , eg. \texttt.

---

### Meta-Review · Program_Chairs · 2021-04-21

**Recommendation:** Reject
**Confidence:** 5

**Metareview:**

The four reviewers agreed that the work presented is interesting and relevant for the workshop. However, there is a clear consense that is not ready for its publication. We would encourage the authors to join us in the W3C CG (https://www.w3.org/community/kg-construct/) and help us in the definition of the next generation of mapping languages taking into account the identified challenges and the proposals presented in this paper (bc they are surely interesting contributions to take into account)! Hope to see the authors as attendees and also new submissions in the next editions of the workshop

David

---

### Decision · Program_Chairs · 2021-04-23

Reject